# Nonvolatile optical phase shift in ferroelectric hafnium zirconium oxide

Kazuma Taki[1], Naoki Sekine[1], Kouhei Watanabe[1], Yuto Miyatake [1], Tomohiro Akazawa[1], Hiroya Sakumoto[1], Kasidit Toprasertpong [1], Shinichi Takagi[1] & Mitsuru Takenaka [1] ✉

A nonvolatile optical phase shifter is a critical component for enabling the fabrication of programmable photonic integrated circuits on a Si photonics platform, facilitating communication, computing, and sensing. Although ferroelectric materials such as $BaTiO_3$ offer nonvolatile optical phase shift capabilities, their compatibility with complementary metal-oxide-semiconductor fabs is limited. $Hf_{0.5}Zr_{0.5}O_2$ is an emerging ferroelectric material, which exhibits complementary metal-oxide-semiconductor compatibility. Although extensively studied for ferroelectric transistors and memories, its application to photonics remains relatively unexplored. Here, we show the optical phase shift induced by ferroelectric $Hf_{0.5}Zr_{0.5}O_2$. We observed a negative change in refractive index at a 1.55 μm wavelength in a pristine device regardless of the direction of the applied electric field. The nonvolatile phase shift was only observed once in a pristine device. This non-reversible phase shift can be attributed to the spontaneous polarization within the $Hf_{0.5}Zr_{0.5}O_2$ film along the external electric field.

Silicon (Si) photonics have experienced remarkable advancements over the past two decades, playing a pivotal role in facilitating the exponential growth of optical fiber communication traffic[1,2]. This progress has led to an exponential increase in the integration capacity of optical elements on a Si photonic integrated circuit (PIC), fueling expectations for the development of programmable PICs, catering to not only optical communication but also computing and sensing applications[3–8]. An essential component in reconfiguring the functionality of a programmable PIC is the optical phase shifter. This component plays a crucial role in adjusting the phase of a light signal as it propagates through an optical waveguide. In the context of a Si programmable PIC, the thermo-optic (TO) phase shifter is the most commonly used type of optical phase shifter. While the TO phase shifter has a simple structure and can be easily integrated, its drawback lies in its large power consumption, which poses challenges for large-scale integration[9]. However, alternative approaches have been explored to address this issue. Low-power and low-loss optical phase shifters have been fabricated using III-V/Si hybrid integration[10],

micro-electro-mechanical systems (MEMS)[11], ferroelectrics[12–15], phase change materials[16,17], 2D materials[18,19], and piezoelectric actuation[20]. Among these alternatives, the optical phase shifter based on ferroelectric $BaTiO_3$ shows promise, particularly owing to its low-loss, nonvolatile operation driven by an external electric field[12,13]. The non-volatile nature of the $BaTiO_3$-based phase shifter allows for zero static power consumption and simplification of electric wiring through a crossbar configuration[21]. Moreover, the non-volatility of a phase shifter is essential for an energy-efficient deep learning accelerator based on a programmable PIC utilizing an in-memory computing architecture, which helps overcome the von Neumann bottleneck[22]. However, a challenge arises with $BaTiO_3$, as it is incompatible with the complementary metal-oxide-semiconductor (CMOS) process, making it difficult to fabricate $BaTiO_3$-based optical phase shifters in CMOS fabs. Hence, there is a strong desire for a CMOS-compatible ferroelectric material.

In this work, we investigate the nonvolatile optical phase shift in orthorhombic $Hf_{0.5}Zr_{0.5}O_2$ (HZO), which was first reported to exhibit

[1]Department of Electrical Engineering and Information Systems, The University of Tokyo, 7-3-1 Hongo, Bunkyo-ku, Tokyo 113-8656, Japan.
✉e-mail: takenaka@mosfet.t.u-tokyo.ac.jp

ferroelectricity in 2011[23–25]. Since $HfO_2$-based ferroelectrics are CMOS-compatible, a wide variety of electron devices such as ferroelectric transistors and ferroelectric memories have been investigated intensively[26,27]. However, the optical properties of $HfO_2$-based ferroelectrics have not been fully explored yet[28–30]. Therefore, it is of utmost importance to evaluate the refractive index change induced by the electric field in ferroelectric HZO, as it has the potential to be applied as a CMOS-compatible Si optical phase shifter.

## Results

### Device structure and fabrication

To evaluate the electric-field-induced refractive index change in HZO, we prepared a SiN optical waveguide surrounded by HZO (10 nm)/$Al_2O_3$ (1 nm) stacks (See Methods and Supplementary Fig. S1a and S1b) with a Mach–Zehnder interferometer (MZI), as shown in Fig. 1a, b. The SiN waveguide operating at a 1.55 μm wavelength was fabricated on a thermally oxidized Si wafer. The ferroelectricity in HZO reaches its maximum when crystalizing at a film thickness of 10 nm[31]. To enhance the overlap between the confined light in the waveguide and the HZO layers, we stacked 10 nm-thick HZO films in three cycles with 1 nm-thick $Al_2O_3$ interlayers by atomic layer deposition (ALD) so that the total ferroelectric thickness achieves 30 nm while each 10 nm-thick

layer is separately crystallized[32]. The $Al_2O_3$ interlayer induces the generation of oxygen vacancies in the HZO layers, thereby promoting the HZO layers to exhibit ferroelectric behavior[33], in conjunction with the stress-induced crystallization of the HZO layers by TiN capping[34]. Note that annealing temperature of 400 °C for achieving ferroelectricity of HZO is much lower than the maximum temperature of the front-end-of-line (FEOL) process for CMOS transistors. Therefore, there are no major difficulties in integrating HZO into the FEOL process, as reported in ref. 35. While we anticipate that increasing the number of cycles will augment the refractive index change and potentially reduce device length, achieving robust ferroelectric behavior might require meticulous process optimization. A bias voltage was applied to generate a horizontal electric field across the waveguide, as shown in Fig. 1b. Figure 1c shows a plan view of the fabricated SiN waveguide devices. We prepared an asymmetric Mach-Zehnder interferometer (AMZI) with 4.5 mm-long phase modulators, enabling us to evaluate the phase shift through the resonance wavelength peak shift in the transmission spectrum of the AMZI. To evaluate the change in optical loss with respect to the application of a bias voltage, we also prepared straight optical phase modulators. We also fabricated samples with a nonferroelectric $HfO_2$ or $SiO_2$ cladding layer as references (See Supplementary Fig. S1c and S1d). The width of the SiN waveguide

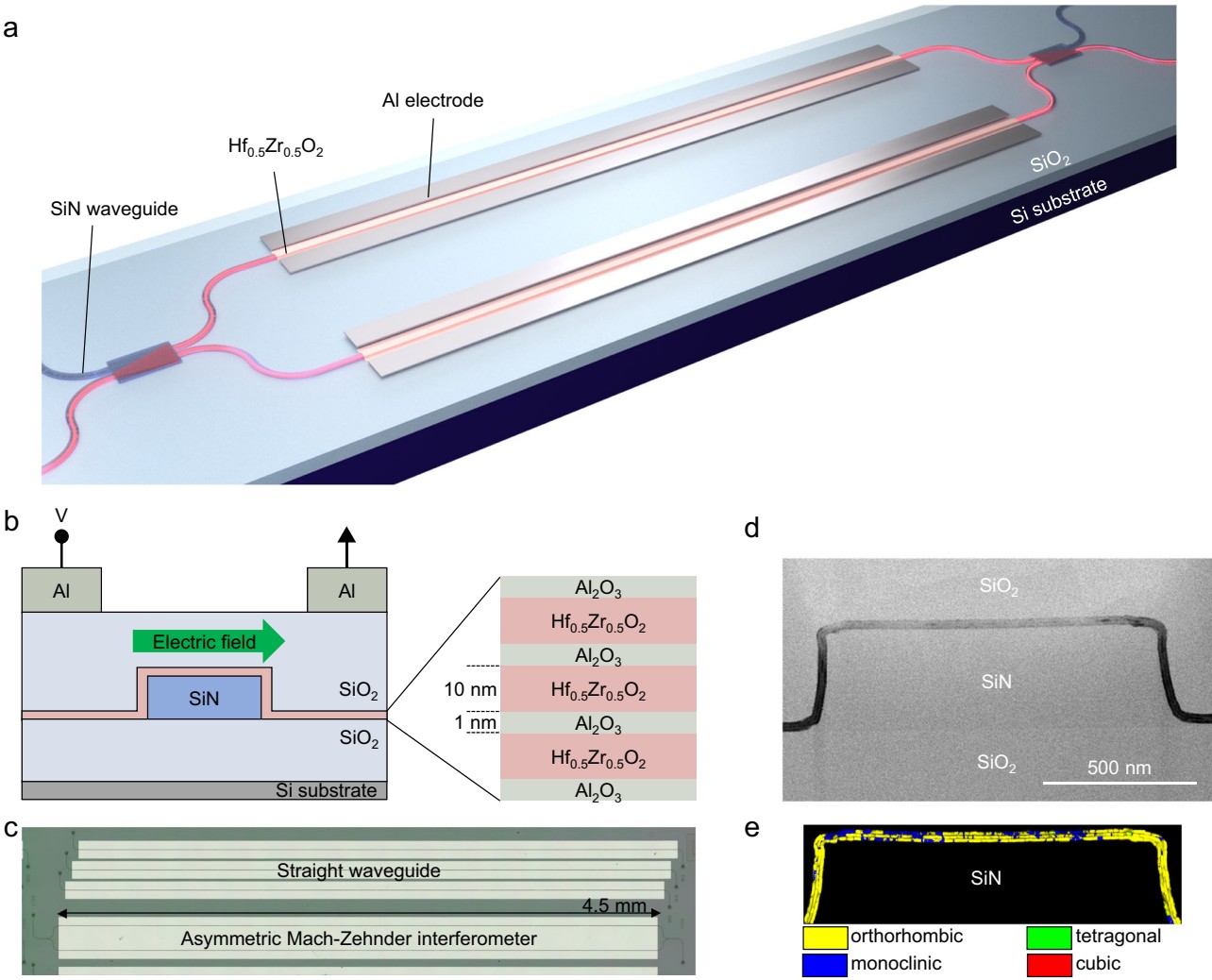

**Fig. 1 | Schematic and images of a SiN waveguide with $Hf_{0.5}Zr_{0.5}O_2$-based optical phase shifter.** **a** 3D view of a Mach-Zehnder interferometer based on a SiN waveguide with $Hf_{0.5}Zr_{0.5}O_2$-based optical phase shifters. Al electrodes are formed for applying an electric field to $Hf_{0.5}Zr_{0.5}O_2$. **b** Cross-sectional structure of the optical phase shifter with $Hf_{0.5}Zr_{0.5}O_2$/$Al_2O_3$ stacks deposited on a SiN waveguide. An external electric field is applied along the transverse direction through the Al electrodes. **c** Plan-view microscopy image of the fabricated device. An AMZI and straight waveguides with grating couplers are prepared for optical phase and loss evaluation. **d** Cross-sectional TEM image of the SiN waveguide with $Hf_{0.5}Zr_{0.5}O_2$/$Al_2O_3$ stacks. **e** Phase map of the $Hf_{0.5}Zr_{0.5}O_2$ layers.

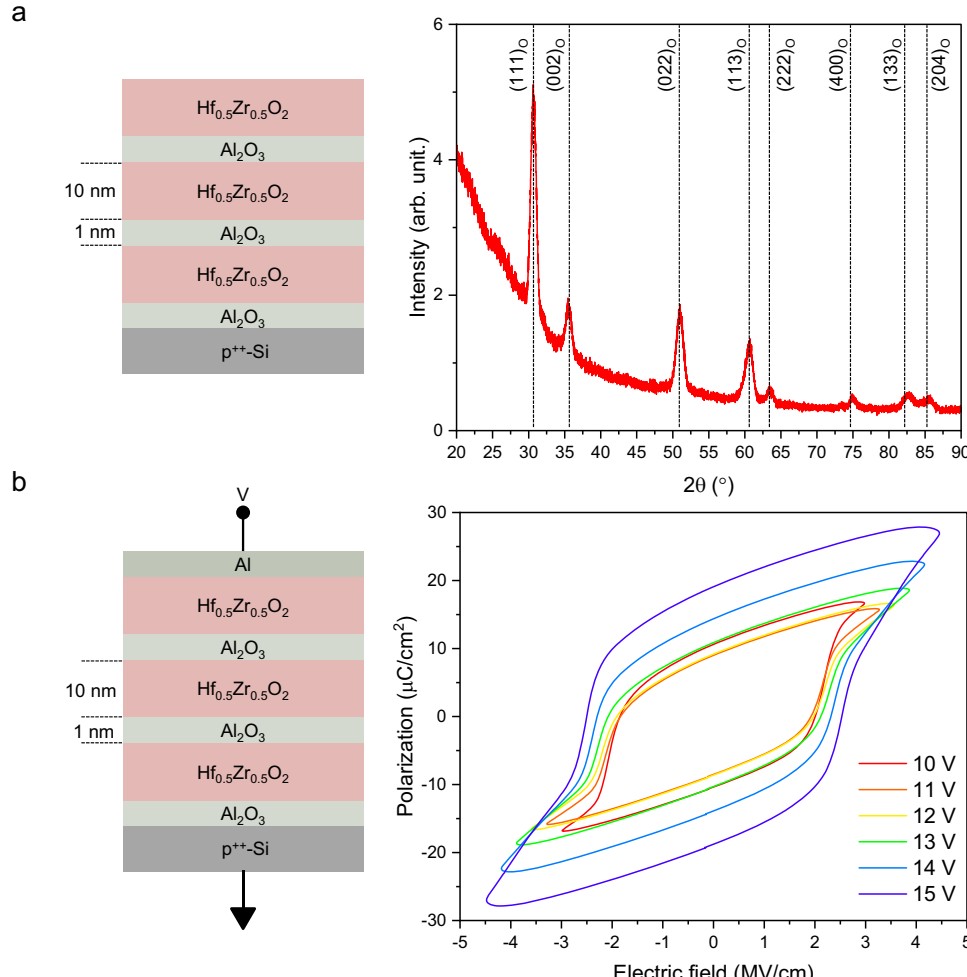

**Fig. 2 | Material characterizations of Hf$_{0.5}$Zr$_{0.5}$O$_2$/Al$_2$O$_3$ stacks. a** Grazing-incidence X-ray diffraction of the Hf$_{0.5}$Zr$_{0.5}$O$_2$/Al$_2$O$_3$ stacks deposited on a Si substrate. The orthorhombic phase of the Hf$_{0.5}$Zr$_{0.5}$O$_2$ layer is confirmed by the diffraction peaks. **b** PV characteristics of the metal−insulator−semiconductor capacitor composed of the Hf$_{0.5}$Zr$_{0.5}$O$_2$/Al$_2$O$_3$ stacks. The hysteresis PV curves clearly indicate the ferroelectricity of the Hf$_{0.5}$Zr$_{0.5}$O$_2$ layer.

was designed to be 1.2 μm for a single-mode operation (See Supplementary Fig. S2). The multimode interference couplers employed in AMZIs and the grating couplers for fiber coupling were designed for the transverse-electrical (TE) mode of the SiN waveguide (See Supplementary Fig. S2e and S2f).

Figure 1d presents a cross-sectional transmission electron microscopy (TEM) image of the SiN waveguide with the conformally deposited HZO/Al$_2$O$_3$ layers. Magnified TEM images clearly revealed three polycrystalline HZO layers separated by Al$_2$O$_3$ interlayers (See Supplementary Fig. S3). Automated crystal orientation mapping in transmission electron microscopy (ACOM-TEM)[36] with a precession electron diffraction angle of 0.5° was conducted to analyze the phase and orientation of the polycrystalline HZO layers, as depicted in Fig. 1e. We found that the majority of the HZO layers on the top and sidewalls of the SiN waveguide were in the orthorhombic crystal phase, which exhibits ferroelectric properties. Since the thickness of each Al$_2$O$_3$ layer is only 1 nm, which is significantly smaller than the operating wavelength, the influence of the presence of the Al$_2$O$_3$ layers on the waveguide property is negligible.

### Ferroelectric properties

First, we evaluated the properties of the HZO/Al$_2$O$_3$ stacks prepared on a Si substrate, as shown in Fig. 2. To evaluate the crystallinity of the poly HZO/Al$_2$O$_3$ stacks, we performed grazing-incidence X-ray diffraction, as shown in Fig. 2a. An orthorhombic phase or rhombohedral

phase is expected for ferroelectric HZO[37]. We found that the diffraction peaks correspond to the orthorhombic phases[25]. Polarization−voltage (PV) characteristics of the metal−insulator−semiconductor capacitor composed of the HZO/Al$_2$O$_3$ stacks in various voltage ranges are shown in Fig. 2b. As expected for the measured orthorhombic phase in Figs. 1d and 2a, a clear hysteresis of PV characteristics was observed, confirming the ferroelectricity of HZO/Al$_2$O$_3$ stacks. When a bias voltage was swept between −15 V and +15 V, the remnant polarization of the HZO/Al$_2$O$_3$ stacks was approximately 20 μC/cm$^2$, comparable to that of a 10 nm-thick HZO single layer[38,39]. By inserting a 1 nm Al$_2$O$_3$ layer for every 10 nm HZO layer[40,41], we obtained good ferroelectricity even in the HZO layer with a total thickness of 30 nm (See Supplementary Fig. S4).

### Optical properties

Next, we measure the transmission of the fabricated waveguide devices to evaluate the refractive index change of the HZO layer in response to an applied voltage. The measurement was conducted by injecting a continuous-wave laser light with wavelengths ranging from 1480 nm to 1580 nm through grating couplers (see Methods). As shown in Supplementary Fig. S5a, S5b, the propagation loss of the SiN waveguide with SiO$_2$ cladding was evaluated to be 6.1 dB/cm. The propagation loss with HZO is nearly identical to that with SiO$_2$, as shown in Supplementary Fig. S5c. The ellipsometry measurements shown in Supplementary Fig. S6 present the refractive index and

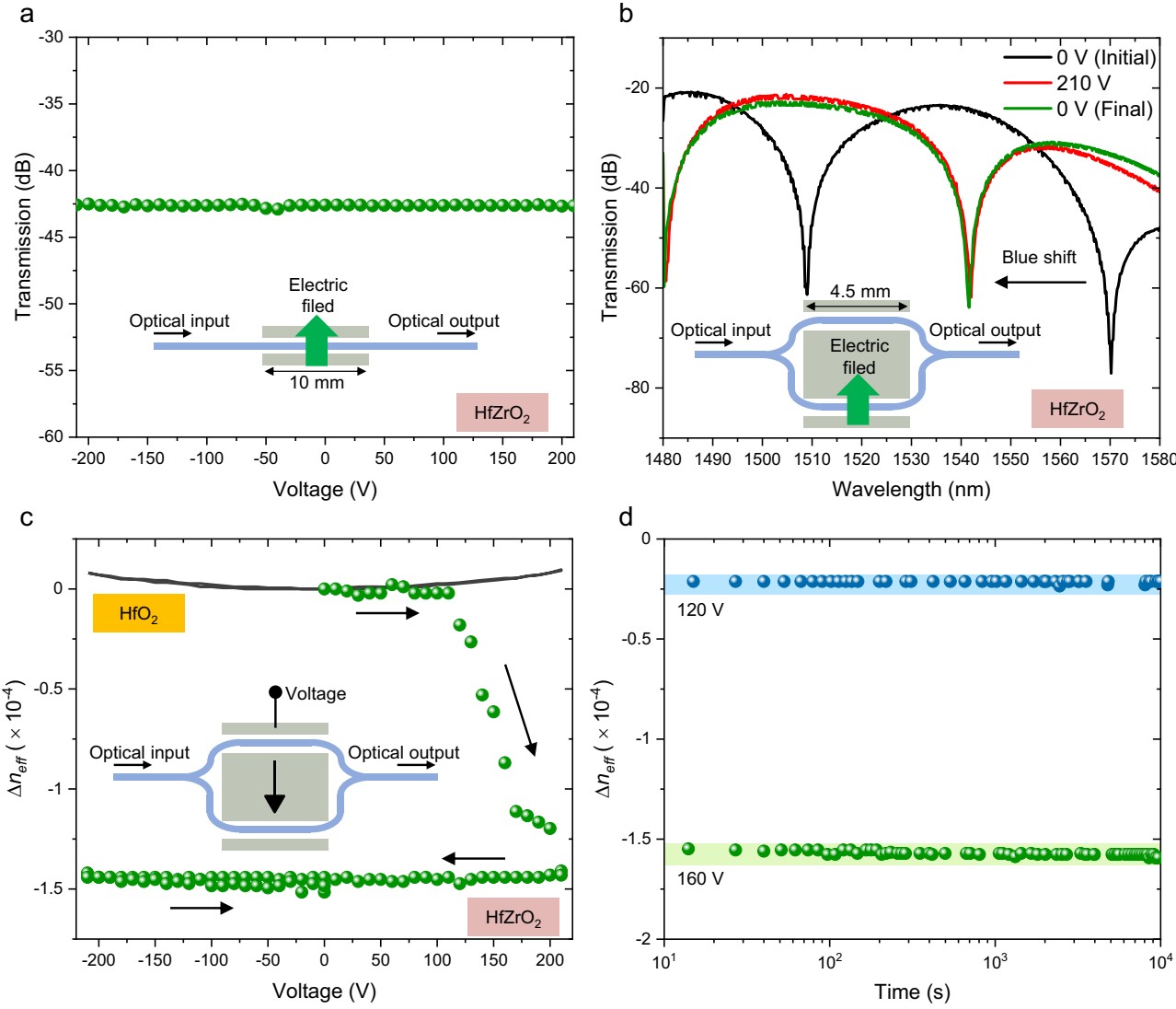

**Fig. 3 | Optical properties of Hf$_{0.5}$Zr$_{0.5}$O$_2$-based optical phase shifter. a** Optical transmission measurement of the 10 mm-long straight device with respect to the voltage applied. No significant intensity modulation was observed, indicating that the free-carrier effects are negligible. **b** Transmission spectra of AMZI with the 4.5 mm-long phase shifter. A blue shift in the resonance wavelength peak is observed when 210 V is applied once, showing the nonvolatile optical phase shift. **c** Change in the effective refractive index when a voltage is swept between +210 and −210 V. When a voltage increases from 0 V to 210, a significant negative refractive index change is observed. Following the initial significant refractive index change, only a small positive change is observed, which can be attributed to the Kerr effect in SiN. A similar Kerr effect is also observed in the HfO$_2$ and SiO$_2$ devices. **d** Retention characteristics of the nonvolatile refractive index change after a voltage application of 120 V or 160 V.

extinction coefficient of a 10 nm-thick HZO film. Given HZO's wide-bandgap nature, ellipsometry at a wavelength of 1.55 μm didn't reveal any distinct extinction coefficient, further supporting the notion of negligible optical loss associated with HZO. Since the TO effect could potentially contribute to the change in the effective refractive index of the propagating light, we measured the leakage current when applying a bias voltage between the two electrodes of the phase shifter. Our findings revealed a leakage current of less than approximately 300 pA with a 200 V bias voltage (See Supplementary Fig. S7). Considering that the TO effect typically requires more than 10 mW to induce a π phase shift in a SiN waveguide[42], we can confidently conclude that the TO effect is negligible in our experiments. We also investigated the impact of free-carrier plasma dispersion and absorption induced by an applied electric field, which is particularly significant in the case of Si. This investigation involved measuring the transmission of a 10 mm-long straight waveguide in response to the bias voltage applied, as shown in Fig. 3a. Notably, we observed no optical intensity modulation

even with the application of a 210 V bias voltage, suggesting that the free-carrier effect in the SiN waveguide is negligible.

Then, we proceeded to directly assess the optical phase shift using the AMZI configuration by applying a bias voltage to one of the MZI arms, as illustrated in Fig. 3b. The transmission spectrum of the AMZI before bias voltage application is shown as the black line in Fig. 3b. Notably, the spectrum exhibits periodic wavelength resonance peaks attributed to the length difference between the two MZI arms. Subsequently, we applied a bias voltage of 210 V to a 4.5 mm-long optical phase shifter within the longer AMZI arm, as indicated by the red line in Fig. 3b. The direction of the applied electric field is illustrated as a green arrow in the inset of Fig. 3b. The Simulation result of the electric field distribution at 200 V is illustrated in Supplementary Fig. S8a. The electric field in the HZO layer atop the SiN waveguide is proportional to an applied voltage, as shown in Supplementary Fig. S8b. The external electric field reaches approximately 0.35 MV/cm at 200 V. The transmission spectrum, in this case, reveals a distinctive blue shift in the

resonance wavelength peak, clearly indicating a negative change in the effective refractive index of the propagating light. Importantly, the blue-shifted wavelength peak remained unchanged even upon returning the bias voltage to 0 V, indicating the attainment of a non-volatile optical phase shift. The change in the effective refractive index of the pristine HZO device in response to an applied voltage is depicted by the green circular symbols in Fig. 3c. In this measurement, a bias voltage was applied to the shorter AMZI arm, as depicted in the inset of Fig. 3c. Although no notable change in the effective refractive index occurred within the range of applied voltages from 0 V to 120 V, a substantial negative change in the effective refractive index became evident when the applied voltage exceeded 120 V. Upon reverting the applied voltage from 210 V back to 0 V, the change in the effective refractive index remained relatively constant, as depicted in Fig. 3b, with a magnitude of approximately $-1.5 \times 10^{-4}$ even at 0 V. In this measurement, the direction of the electric field is opposite to that in Fig. 3b, yet the refractive index change remains negative, as similarly observed in Fig. 3b. Therefore, the initial substantial change in the refractive index is consistently negative, irrespective of the electric field direction. Continuing to sweep the voltage down to −210 V did not result in the same large change in the effective refractive index as initially observed. Note that the $HfO_2$ and $SiO_2$ reference devices exhibited a positive change in the effective refractive index, which is approximately 10 times smaller than that in the HZO device, as shown by the black line in Fig. 3c and in Supplementary Fig. S9. The refractive index change increased with the square of the voltage, which can be attributed to the Kerr effect in SiN[43]. From this comparison, we can deduce that the substantial negative change in the effective refractive index originates from the ferroelectric HZO layer. Since the nonvolatile phase shift was observed only when a voltage was applied to the pristine sample as shown in Fig. 3c, the presented phase shifter works as a one-time memory. Consequently, a wake-up process might not yield any significant benefits. Supplementary Fig. S10 illustrates the refractive index change in a couple of the samples. As depicted in this figure, we observed a basically similar one-time unidirectional phase shift, indicating the reproducibility of our findings. The variations in the amount of refractive index change and the voltage at which the refractive index change begins could be attributed to variations in device fabrication. The retention characteristics of the nonvolatile refractive index changes observed in the HZO sample were then evaluated. Figure 3d shows the results of measuring the change in refractive index over time after a voltage of 120 V or 160 V was applied and then the voltage was returned to 0 V again. It can be seen that there is no significant change in the refractive index throughout measurements up to $10^4$ s. This retention time is comparable to that of the $BaTiO_3$-based optical phase shifter[12]. We expect a longer retention time as reported in ref. 44. As shown in Fig. 3c, the refractive index remains nearly unchanged when the applied voltage is reverted to 0 V. This suggests the potential to achieve a minimum of 10 intermediate states, as evidenced by the measured points between 110 V and 210 V in Fig. 3c. Therefore, we can realize a multilevel nonvolatile optical phase shifter using ferroelectric HZO. Measuring the small change in optical loss during the retention measurement is challenging due to the significant impact of optical fiber alignment changes on the loss measurement for optical input and output. The retention is dominated by polarization change in HZO. Since we observed no change in optical loss during the polarization change induced by applying voltage, as shown in Fig. 3a, it is reasonable to expect a negligible change in optical loss during the retention measurement.

## Discussion

To delve into the physical mechanism underlying the nonvolatile optical phase shift in ferroelectric HZO, we conducted ACOM-TEM of the HZO layer. This analysis enabled us to explore the alteration in the orientation of the orthorhombic phase induced by the application of an electrical field. Figure 4a illustrates a grain map of the orthorhombic HZO layer on the SiN waveguide before the application of any electrical field. In Fig. 4b, an inverse pole figure of the HZO layer along the direction of the electrical field shown in Fig. 1a, defined as the transverse direction, is presented; here, the (001) orientation is defined as the polarization axis of an orthorhombic HZO crystal. Figure 4c, d show a grain map and an inverse pole figure of the HZO layer, respectively, after the application of 210 V. From the comparison of Fig. 4b, d, it becomes evident that the polarization axis shifts more toward the transverse direction owing to the applied voltage[45,46]. Figure 4e shows the product of the normalized distribution density of the orientation angle θ of the orthorhombic phase and cosθ, which is regarded as the component contributing to the refractive index change. Here, θ is defined as the angle between the polarization axis and the transverse direction. It is clearly seen that the product-sum of the normalized distribution density and cosθ increased by approximately 30% after the voltage application. We observed a similar trend in the HZO layer at the sidewall of the SiN waveguide, as shown in Supplementary Fig. S11.

Taking into account the rotation of the polarization axis of HZO along the direction of the applied electric field, as observed in Fig. 4, the negative change in the refractive index, as depicted in Fig. 3c, regardless of the electric field direction, can be elucidated using two physical models. One such model is the quadratic electro-optic effect induced by the spontaneous polarization of HZO[47–49]. As observed in $LiNbO_3$, the refractive index reduction is proportional to the square of the spontaneous polarization through the quadratic electro-optic effect in ferroelectric materials. When a voltage is applied to a pristine device, the spontaneous polarization is expected to evolve along the transverse direction, as depicted in Fig. 4. As a result, we expect to observe a negative refractive index change that is independent of the direction of voltage application. With the TE-mode light predominantly confined in the SiN core, the optical confinement factor within the HZO layer stands at approximately 4.5%. Taking the quadratic electro-optic coefficient of HZO as the same value as that of $LiNbO_3$ ($0.09 \, m^4C^{-2}$)[49], and considering a spontaneous polarization of $10 \, \mu C/cm^2$ and an optical confinement factor within the HZO layer, the estimated effective refractive index change stands at approximately $-1.8 \times 10^{-4}$, which is close to the experimental value shown in Fig. 3c. Hence, the quadratic electro-optic effect induced by the spontaneous polarization in HZO after the first voltage sweep presents a plausible explanation for the nonvolatile optical phase shift. During the second voltage sweep, the spontaneous polarization in HZO remains undisturbed, while an external electric field modulates only the para-electric component of the polarization in HZO. As shown in Supplementary Fig. S8, the external electric field at 200 V is approximately 0.35 MV/cm. With this electric field and a dielectric constant of 30, the polarization from the paraelectric component is calculated to be approximately $0.9 \, \mu C/cm^2$, which is approximately 10 times smaller than a spontaneous polarization of $10 \, \mu C/cm^2$. The refractive index change caused by the quadratic electro-optic effect is proportional to the square of the polarization[49]. Therefore, the refractive index change by the paraelectric polarization through the quadratic electro-optic effect in HZO, resulting from an external electric field in the second voltage sweep, is expected to be roughly 100 times smaller than the nonvolatile refractive index change observed in the first voltage sweep due to the spontaneous polarization in HZO. Thus, the refractive index change induced by an external electric field through the quadratic electro-optic effect of the paraelectric polarization in HZO remains unobservable due to the presence of the quadratic electro-optic effect in SiN.

The other model pertains to the birefringence of HZO. According to numerical predictions, the refractive index of orthorhombic $HfO_2$ along the polarization axis is 2.081 at a wavelength of 633 nm, whereas the refractive indices along the other two axes are expected to be

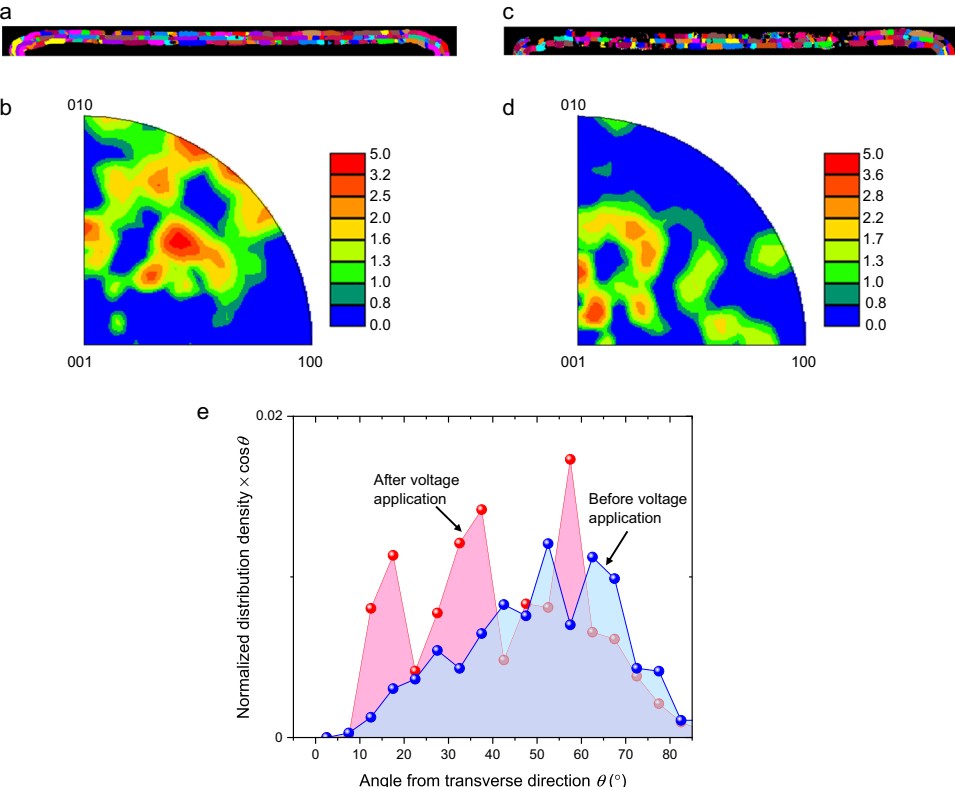

**Fig. 4 | Automated crystal orientation mapping of the $Hf_{0.5}Zr_{0.5}O_2$ layers on the top of the SiN waveguide. a** Grain map and (**b**) inverse pole figure of the orthorhombic $Hf_{0.5}Zr_{0.5}O_2$ layer before voltage application. **c** Grain map and (**d**) inverse pole figure of the orthorhombic $Hf_{0.5}Zr_{0.5}O_2$ layer after voltage application. **e** Product of the normalized distribution density of the orientation angle $\theta$ of the orthorhombic phase and $\cos\theta$. The polarization axis rotates toward the direction of an external electric field owing to the voltage application, which contributes to the change in refractive index.

2.059 and 2.122[30]. Even at a wavelength of 1550 nm, the refractive indices are expected to be close to that at 633 nm[50]. In directions perpendicular to the polarization axis, HZO crystals are assumed to be randomly oriented. Consider the scenario in which the polarization axis is oriented perpendicular to the transverse direction. Since the measurement in Fig. 3 employs TE-mode light, in which the optical electric field is predominantly polarized along the transverse direction, the refractive index for the propagating light becomes the average of the refractive index perpendicular to the polarization axis, i.e., 2.0905. Upon rotating the polarization axis along the transverse direction through voltage application, the refractive index for the propagating light changes to that along the polarization axis, i.e., 2.081. Considering the optical confinement factor within the HZO layer, the maximum estimated change in effective refractive index reaches approximately $-4.3 \times 10^{-4}$, which is approximately three times higher than the experimental value depicted in Fig. 3c. This discrepancy indicates that the polarization axis does not completely rotate toward the direction of the external electric field due to insufficient field strength. The same logic applies to $ZrO_2$. Consequently, the refractive index reduction observed regardless of the direction of voltage application can also be explained by the birefringence of orthorhombic HZO.

In conclusion, we successfully demonstrated the nonvolatile optical phase shift induced by applying a voltage to ferroelectric HZO deposited on a SiN waveguide. This observed phenomenon can be attributed to the rotation of the polarization axis of orthorhombic HZO along the direction of the external electric field. Notably, the nonvolatile phase shift was only observed once in a pristine HZO sample. Additionally, the consistent negative refractive index change, regardless of the electric field direction, signifies unidirectional phase shifting. Although such a one-time unidirectional phase shift may indeed have limited applications, our discovery of the nonvolatile

optical phase shift observed in HZO remains valuable for one-time optical memory applications in programmable PICs. For instance, it can be utilized to store the weights of a matrix for multiply-accumulation operations. This is particularly relevant because the weights for inference purposes remain unchanged once the learning process is completed. Another potential application of our discovery is the correction of initial phase errors in programmable PICs. Given that initial phase errors are inevitable due to fabrication variations, the one-time unidirectional phase shifter proves to be useful in this regard. Presently, owing to limitations in the measurement equipment, applying a voltage higher than 210 V is not feasible. Nevertheless, it is anticipated that employing a larger electric field for HZO could yield an even more substantial optical phase shift. Since the leakage current was less than 300 pA even at 200 V, the switching energy could be very small even with a slow switching time. For instance, assuming the switching speed of 10 s at 200 V, the estimated switching energy amounts to approximately 0.6 μJ. This level of energy consumption might not be significant for the reconfiguration purpose of PICs. In this study, no linear electro-optic effect, i.e., the Pockels effect, was observed in HZO. However, there is a possibility of observing the Pockels effect in HZO by applying a larger electric field using a slot waveguide[51] or plasmonic waveguide[52]. For instance, the device structure of the Si slot waveguide with HZO in the slot gap is illustrated in Supplementary Fig. S12a. The simulated electric field distribution at 20 V in Supplementary Fig. S12b shows that the external electric field is concentrated in the slot gap. As shown in Supplementary Fig. S12c, the slot device, with bias voltages exceeding 10 V, allows an external electric field of 1 MV/cm, which is close to the coercive electric field of HZO[53]. Since ALD facilitates the conformal deposition of HZO, it allows HZO to be deposited on both sidewalls of the slot. Consequently, the HZO film grows laterally from the sidewall. In this scenario, a

50 nm-thick HZO layer is necessary on one side to fill the 100 nm-wide gap of the slot. Such a thickness is realistic for maintaining the ferroelectric behavior of HZO. The utilization of a Si slot waveguide also facilitates a reduction in the switching time of the nonvolatile optical phase shift observed in this study. Based on transient measurements shown in Supplementary Fig. S13a and S13b, the observed switching speed appears slow, ranging from 10 to 100 s due to an insufficient electric field in the SiN waveguide device. Referring to the nucleation-dominated switching model in HZO[54], the switching time follows an exponential reduction relative to the square of the external electrical field. Extrapolating from the measured switching speeds, a switch time of less than 1 μs could be achieved with an external electric field of approximately 0.7 MV/cm, as shown in Supplementary Fig. S13c. As mentioned earlier, such a field strength can be attained using the slot device. Hence, the utilization of a slot waveguide or plasmonic waveguide demonstrates potential in achieving switching times below 1 μs. We anticipate that our findings will trigger and promote the study of optical nonlinear effects in HZO, leading to the progress of Si PICs employing ferroelectric HZO and making significant contributions to the evolution of communication, computing, and sensing.

## Methods

The SiN waveguide with the HZO/Al$_2$O$_3$ stack was fabricated as follows (See Supplementary Fig. S1 for the detailed fabrication procedure). First, a 330 nm-thick SiN layer was deposited on a thermally oxidized Si wafer by plasma–enhanced chemical vapor deposition (PECVD). The thickness of the thermally oxidized SiO$_2$ layer was 4 μm. SiN waveguides were fabricated by electron-beam (EB) lithography and inductively coupled plasma (ICP) dry etching. Three HZO/Al$_2$O$_3$ multilayer stacks were then deposited on the Si waveguide by ALD. After depositing TiN on the HZO/Al$_2$O$_3$ stacks, annealing at 400 °C for 1 min was performed to achieve the ferroelectricity of HZO. TiN was removed by wet etching, and SiO$_2$ passivation was carried out by PECVD. The thickness of the SiO$_2$ cladding was 600 nm. Finally, Al electrodes were formed by metal sputtering and lift-off. For optical intensity and phase modulation measurements, a continuous-wave light from a tunable laser source (Santec, TSL-510) was injected into the SiN waveguide from a single-mode optical fiber through a SiN grating coupler. The polarization of the input light was tuned to the TE mode of the SiN waveguide using an in-line polarization controller. The output from the SiN waveguide was coupled back to the optical fiber through a grating coupler and measured by an InGaAs power meter (Agilent, 81624 A). A precision source/measure unit (Agilent, B2902A) was used for voltage application and current measurement.

## Data availability

The data that support the findings of this study have been included in the main text and Supplementary Information. All other relevant data supporting the findings of this study are available from the corresponding authors upon request.

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

## Acknowledgements

This work was partly based on results obtained from projects (JPNP16007 received by M.T.) commissioned by the New Energy and Industrial Technology Development Organization (NEDO) and partly supported by JST, CREST (JPMJCR2004 received by M.T.), JST, MIRAI (JPMJMI20A1 received by M.T.), and "Advanced Research Infrastructure for Materials and Nanotechnology in Japan (ARIM)" of the Ministry of Education, Culture, Sports, Science and Technology (MEXT) (JPMXP1222UT1028).

## Author contributions

K.Ta. contributed to fabrication, measurement, and manuscript preparation, N.S. contributed to ideas, K.W. contributed to fabrication and measurement, Y.M. contributed to design and measurement, T.A. contributed to the simulation and manuscript preparation, and H.S. contributed to the simulation. K.To and S.T. contributed to the overall discussion. M.T. contributed to ideas and to discussion, and manuscript revision and also provided high-level project supervision.

## Competing interests

The authors declare no competing interests.
