## [Peer Review File · Nature Communications]

Nonvolatile optical phase shift in ferroelectric hafnium zirconium oxideEditorial Note: This manuscript has been previously reviewed at another journal that is not operating a transparent peer review scheme. This document only contains reviewer comments and rebuttal letters for versions considered at *Nature Communications*.

REVIEWER COMMENTS

Reviewer #1 (Remarks to the Author):

The manuscript has been revised and the primary concerns have been addressed. However, the response to comment #4 is still not clear for me. The new calculations indicate that the measured refractive index change (Fig. 3c) is no longer consistent with the hypothesis of the quadratic electro optic effect in HZO as the spontaneous polarization is 10 times reduced at 200 V. If this is case the revised paragraph should be rewritten as it says first that “Hence, the quadratic electro optic effect model presents a plausible explanation for the nonvolatile optical phase shift” and later “Thus, the refractive index change induced by an external electric field through the quadratic electro-optic effect in HZO remains unobservable due to the presence of the quadratic optic effect in SiN”. In other words, the new text could be confusing for readers.

I have also some minor comments:

- The fact that the achieved phase shift is non-reversible should be clearly stated in the abstract.
- Figure 3d should only be shown up to 10^4 as having only one point close to 10^5 is not rigorous enough.
- In the Extended Data Fig. 7, there is a mistake in the leng

We would like to express our gratitude to the reviewers for their diligent evaluation of our manuscript. We sincerely appreciate the reviewer’s valuable comments and insights on our research. According to the review comments, we have revised our paper. We believe that these revisions make our paper appropriate for publication in Nature communications. In the revised manuscript, we have indicated the modified sections by highlighting them in green.

Reviewers' Comments:

#####

Referee #1 (Remarks to the Author):

Comment #1

The manuscript has been revised and the primary concerns have been addressed. However, the response to comment #4 is still not clear for me. The new calculations indicate that the measured refractive index change (Fig. 3c) is no longer consistent with the hypothesis of the quadratic electro optic effect in HZO as the spontaneous polarization is 10 times reduced at 200 V. If this is case the revised paragraph should be rewritten as it says first that “Hence, the quadratic electro optic effect model presents a plausible explanation for the nonvolatile optical phase shift” and later “Thus, the refractive index change induced by an external electric field through the quadratic electro-optic effect in HZO remains unobservable due to the presence of the quadratic optic effect in SiN”. In other words, the new text could be confusing for readers.

Response:

We apologize for the confusing description. During the first voltage sweep, the spontaneous polarization in HZO ($10 \mu\text{C}/\text{cm}^2$) is induced, leading to the nonvolatile optical phase shift through the quadratic electro-optic effect. In the second volage sweep, the spontaneous polarization remains undisturbed while an external electric field modulates only the paraelectric component of the polarization ($0.9 \mu\text{C}/\text{cm}^2$) in HZO. The refractive index change caused by the quadratic electro-optic effect is proportional to the square of the polarization. Therefore, the refractive index change by the paraelectric component of the polarization, resulting from an external electric field in the second voltage sweep, is expected to be roughly 100 times smaller than the nonvolatile refractive index change observed in the first voltage sweep due to the spontaneous polarization in HZO. Hence, our calculation provides a consistent analysis: the nonvolatile optical phase shift caused by the spontaneous polarization in HZO during the first voltage sweep is approximately 10 times greater than the quadratic electro-optic effect in SiN, while, the optical phase shift resulting from the paraelectric polarization

during the second sweep is approximately 10 times smaller than the quadratic electro-optic effect in SiN.

To clarify this point, we have revised manuscript as below.

In the “Discussion and Outlook” section (p. 11):

Original:

As a result, we expect to observe a negative refractive index change that is independent of the direction of voltage application. With the TE-mode light predominantly confined in the SiN core, the optical confinement factor within the HZO layer stands at approximately 4.5%. Taking the quadratic electro-optic coefficient of HZO as the same value as that of LiNbO₃ ($0.09 \text{ m}^4\text{C}^{-2}$)⁴⁹, and considering a spontaneous polarization of $10 \text{ }\mu\text{C}/\text{cm}^2$ and an optical confinement factor within the HZO layer, the estimated effective refractive index change stands at approximately -1.8×10^{-4} , which is close to the experimental value shown in Fig. 3c. Hence, the quadratic electro-optic effect model presents a plausible explanation for the nonvolatile optical phase shift. As shown in Extended Data Fig. 8, the external electric field at 200 V is approximately 0.35 MV/cm. With this electric field, the polarization, assuming a paramagnetic dielectric with a dielectric constant of 30, is calculated to be approximately $0.9 \text{ }\mu\text{C}/\text{cm}^2$, which is approximately 10 times smaller than a spontaneous polarization of $10 \text{ }\mu\text{C}/\text{cm}^2$. Consequently, the refractive index change induced by an external electric field through the quadratic electro-optic effect in HZO is expected to be roughly 100 times smaller than the nonvolatile refractive index change in HZO. Thus, the refractive index change induced by an external electric field through the quadratic electro-optic effect in HZO remains unobservable due to the presence of the quadratic electro-optic effect in SiN.

Revised:

As a result, we expect to observe a negative refractive index change that is independent of the direction of voltage application. With the TE-mode light predominantly confined in the SiN core, the optical confinement factor within the HZO layer stands at approximately 4.5%. Taking the quadratic electro-optic coefficient of HZO as the same value as that of LiNbO₃ ($0.09 \text{ m}^4\text{C}^{-2}$)⁴⁹, and considering a spontaneous polarization of $10 \text{ }\mu\text{C}/\text{cm}^2$ and an optical confinement factor within the HZO layer, the estimated effective refractive index change stands at approximately -1.8×10^{-4} , which is close to the experimental value shown in Fig. 3c. Hence, the quadratic electro-optic effect induced by the spontaneous polarization in HZO after the first voltage sweep presents a plausible explanation for the nonvolatile optical phase shift. During the second voltage sweep, the spontaneous polarization in HZO remains undisturbed, while an external electric field modulates only the paraelectric component of the polarization in HZO. As shown in Extended Data Fig. 8, the external electric field at 200 V is approximately 0.35 MV/cm. With this electric field and a dielectric constant of 30, the polarization from the paraelectric component is calculated to be approximately $0.9 \text{ }\mu\text{C}/\text{cm}^2$, which is approximately 10 times smaller than a spontaneous polarization of $10 \text{ }\mu\text{C}/\text{cm}^2$. The refractive index change caused by

the quadratic electro-optic effect is proportional to the square of the polarization⁴⁹. Therefore, the refractive index change by the paraelectric polarization through the quadratic electro-optic effect in HZO, resulting from an external electric field in the second voltage sweep, is expected to be roughly 100 times smaller than the nonvolatile refractive index change observed in the first voltage sweep due to the spontaneous polarization in HZO. Thus, the refractive index change induced by an external electric field through the quadratic electro-optic effect of the paraelectric polarization in HZO remains unobservable due to the presence of the quadratic electro-optic effect in SiN.

Comment #2

The fact that the achieved phase shift is non-reversible should be clearly stated in the abstract.

Response:

Thank you for the comment. We have modified the abstract to clarify the non-reversible phase shift.

In the “Abstract” section (p. 1):

Original:

The nonvolatile multilevel optical phase shift was confirmed with a persistence of $> 10^4$ s. This phase shift can be attributed to the spontaneous polarization within the HZO film along the external electric field.

Revised:

The nonvolatile multilevel optical phase shift was confirmed with a persistence of $> 10^4$ s. **The nonvolatile phase shift was only observed once in a pristine device.** This non-reversible phase shift can be attributed to the spontaneous polarization within the HZO film along the external electric field.

Comment #3

Figure 3d should only be shown up to 10^4 as having only one point close to 10^5 is not rigorous enough.

Response:

Thank you for the suggestion. We have modified Fig. 3d with the time scale up to 10^4 s.

In the “Optical properties” section (p. 9):

Original:

It can be seen that there is no significant change in the refractive index throughout measurements up to 10^4 – 10^5 s. This retention time is comparable to that of the BaTiO₃-based optical phase shifter². We expect a longer retention time as reported in Ref. 44.

Fig. 3d

Revised:

It can be seen that there is no significant change in the refractive index throughout measurements up to 10^4 s. This retention time is comparable to that of the BaTiO₃-based optical phase shifter². We expect a longer retention time as reported in Ref. 44.

Fig. 3d

Comment #4

In the Extended Data Fig. 7, there is a mistake in the leng.

Response:

Thank you for pointing out the typo. We have corrected this typo.

In the caption of Extended Data Fig. 7:

Original:

Extended Data Fig. 7. **Current–voltage curve of 10-nm-long optical phase shifter.** The leakage current is less than 300 pA even at 200 V, suggesting that there is no significant TO effect during the measurement of the phase shift.

Revised:

Extended Data Fig. 7. **Current–voltage curve of 10-mm-long optical phase shifter.** The leakage current is less than 300 pA even at 200 V, suggesting that there is no significant TO effect during the measurement of the phase shift.

REVIEWERS' COMMENTS

Reviewer #1 (Remarks to the Author):

All comments have been conveniently addressed so I recommend the publication of the manuscript.